



# An Integrated Marine Data Collection for the German Bight – Part I:  Subaqueous Geomorphology and Surface Sedimentology

Julian Sievers[1], Peter Milbradt[1], Romina Ihde[3], Jennifer Valerius[2], Robert Hagen[3], Andreas Plüß[3]

[1]smile consult GmbH, Hanover, 30159, Germany
[2]Federal Maritime and Hydrographic Agency of Germany, Hamburg, 20359, Germany
[3]Federal Waterways Engineering and Research Institute, Hamburg, 22559, Germany

*Correspondence to* Julian Sievers (sievers@smileconsult.de, ORCID: 0000-0001-9957-0799)

**Abstract.** The German Bight located within the central North Sea is a hydro- and morphodynamically highly complex system of estuaries, barrier islands and part of the world's largest coherent tidal flats, the Wadden Sea. To identify and understand challenges faced by coastal stakeholders, such as harbor operators or governmental agencies, to maintain waterways and employ numerical models for further analyses, it is imperative to have a consistent data base for both bathymetry and surface sedimentology. Current commercial and public data products are insufficient in spatial and temporal 15 resolution and coverage for recent analyses method. Thus, this first part of a two-part publication series of the German joint project EasyGSH-DB describes annual bathymetric digital terrain models in a 10 m gridded resolution for the German North Sea coast and German Bight from 1996 to 2016 (Sievers et al., 2020a, 10.48437/02.2020.K2.7000.0001), as well as surface sedimentological models of discretized cumulative grain size distribution functions for 1996, 2006 and 2016 on 100 m grids (Sievers et al., 2020b, 10.48437/02.2020.K2.7000.0005). Furthermore, basic morphodynamic and sedimentological 20 processing analyses, such as the estimation of e.g. bathymetric stability or surface maps of sedimentological parameters, are provided (Sievers et al., 2020a, 2020b, see respective download links).



## 1 Introduction

The German Bight and the adjacent central North Sea (Figure 1) are highly complex systems of barrier islands, the world's largest coherent system of coastal tidal flats and multiple estuaries (Elias et al., 2012; Kabat et al., 2012; Benninghof and

Winter, 2019). Recent research interest frequently focuses on the morpho- and hydrodynamic processes in this area, e.g. Elias et al. (2012), Zijl et al. (2013), Heyer and Schrottke (2015), Wang et al. (2015) and Benninghof and Winter (2019). For this, numerical models of various implementations require input parameters gained from e.g. bathymetric or sedimentological data sets to gain further insight and understand longer term processes (Zijl et al., 2013; Arns et al., 2015; Wang et al., 2018; Rasquin et al., 2020) such as the sea level rise. Harbor operators for small- and large-scale maritime

economy and tourism and other marine actors need to be able to identify, understand and potentially counteract changes and developments to the coastline and estuaries and its surface sedimentology to maintain and operate within a profitable margin as well as be prepared for potential hazardous situations (Roeland and Piet, 1995; Kirichek et al., 2018; Wölfl et al., 2019; Kiricheck et al., 2020). Additionally, further applications that are not directly obvious are dependent on bathymetric information, such as search efforts in open-sea scenarios (Wölfl et al., 2019). It is thus vital to have a consistent set of high-

resolution elevation and surface sediment information.

Currently available data sets are still highly volatile in spatial and temporal resolution and coverage are compiled in publicly accessible portals and services (Wölfl et al., 2019). Portals and services such as EMODnet Bathymetry ([www.emodnet-bathymetry.eu](http://www.emodnet-bathymetry.eu)), GEBCO ([www.gebco.net](http://www.gebco.net)) or ETOPO ([www.ngdc.noaa.gov](http://www.ngdc.noaa.gov)) offer bathymetric data sets with spatial resolutions of 70 m to approximately 2 km, depending on the overall spatial coverage, yet in part no certain indication about

important quality elements, such as currentness or timeliness and traceability or data source, of the utilized data set. Surface sedimentological information of the central North Sea are coarse for the application in numerical models with spatial resolutions of approximately 2 to 13 km with predetermined classifications and analyses such as median grain diameter already performed and provided by e.g. Helmholtz Zentrum Geesthacht ([coastmap.hzg.de](http://coastmap.hzg.de)), EMODnet Geology ([www.emodnet-geology.eu](http://www.emodnet-geology.eu)) or Wilson et al. (2018), which hinders custom extraction of parameters such as sorting or

skewness coefficients or grain size classes that for numerical models. Portals for individual data publication, such as Pangaea ([www.pangaea.de](http://www.pangaea.de)), usually offer local data sets at a high resolution. However, a low overall coverage results in a decrease of larger scale scientific value, as region-specific availability and consistency issues with adjacent data exist. Apart from spatial resolution and coverage issues, temporal information is in part inconsistent or unavailable. For hydro- and morphodynamically highly dynamic regions such as the German North Sea coast, a specific date or time span of validity is

essential to adequately analyze the occurring processes. German joint project AufMod began the process of alleviating these issues with both increased temporal and spatial coverage and resolution (Heyer and Schrottke, 2015). Bathymetric regular 50 m gridded digital terrain models (DTMs) were provided from 1982 to 2012 for the inner German Bight as well as surface sedimentological information including full discretized cumulative grain size distribution (GSD) functions for individual further processing in a 250 m grid resolution (Milbradt et al., 2015). However, state-of-the-art modeling studies deploy





horizontal grids and meshes for bathymetry and sedimentology that require an even higher resolution than AufMods initial steps (Zijl et al., 2013; Kösters and Winter, 2014; Hagen et al., 2019; Rasquin et al., 2020; Hagen et al., 2021, in review). With AufMod ending in 2012, the data coverage is outdated.

This paper, as part one of a two-part publication, introduces an integrated data collection created in the German project EasyGSH-DB, which includes geomorphological and surface sedimentological products and analyses in a higher temporal
and spatial resolution than previous products: 21 annual bathymetric terrain models spanning 1996 to 2016 as a 10 m regular grid and 3 surface sediment models of GSD functions valid for 1996, 2006 and 2016 on a 100 m grid are presented for the German Bight (Figure 1). Additionally, the larger German Exlusive Economic Zone (AWZ) is covered in a 100 m bathymetric and 250 m surface sedimentological grid of continuous GSD functions. Additionally, basic further processing steps, such as generation of elevation isobaths, bathymetric analyses and sedimentological maps and parameters, are
provided as well.

Part two of this publication (Hagen et al., 2021, in review) displays hydrodynamic numerical modeling results based on the base data presented hereafter. All data products are available for free download, if applicable, provided with additional meta data for quality and usability assessment.

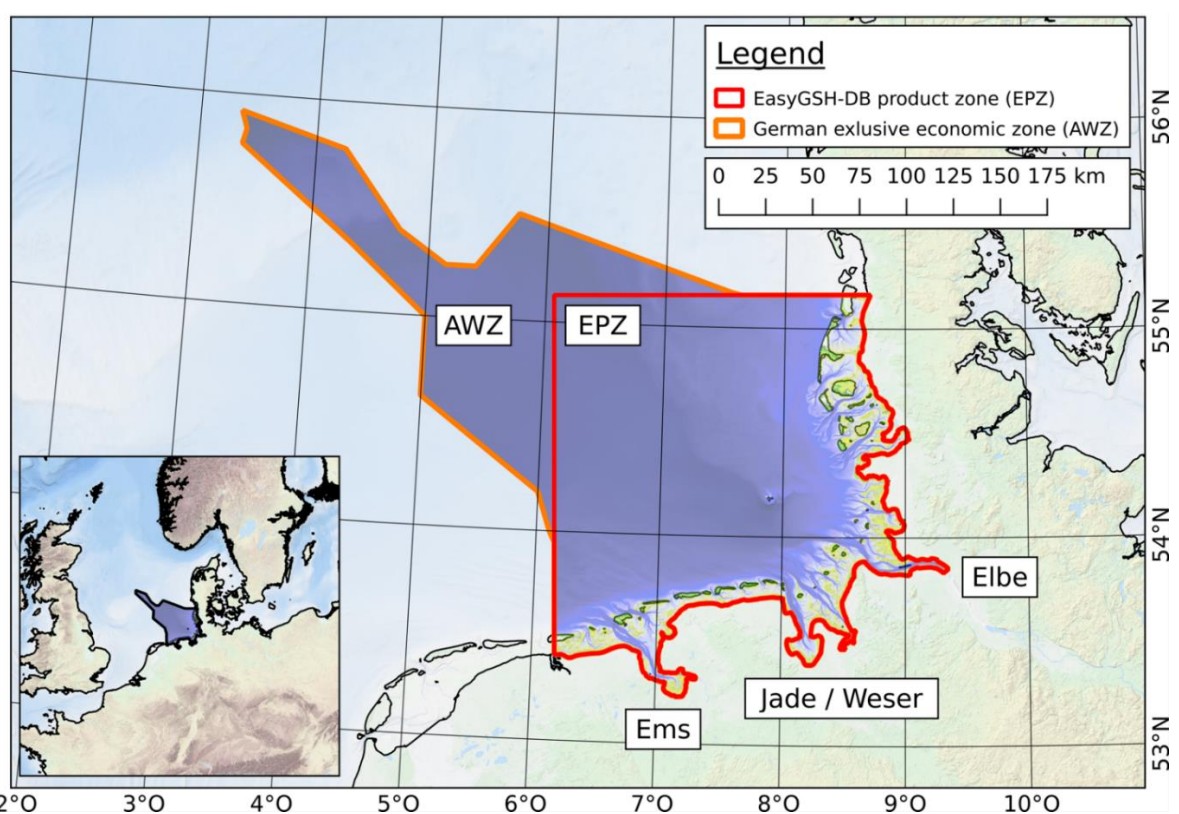

**Figure 1: Spatial extent of the German Bight with EasyGSH-DBs product zone (EPZ) and larger German Exclusive Economic Zone (AWZ) within the North Sea, with a bathymetric model of 1996. Positions of major estuaries given for reference.**

## 2 The Methodological Framework of the Functional Seabed Model (FSM)

Usually, the only way to obtain data valid for a specific date is to measure it at that specific point in time. Studies to undertake surveys in a very high, quasi-continuous, temporal resolution alleviate this issue, they are, however, generally very small scale or on singular transects (Gallagher et al., 1998; Moulton et al., 2014). Concepts to temporally interpolate hydrographic and bathymetric information between fewer temporal sample points to reduce time and costs of such studies are well known (Moulton et al., 2014; Kuusela et al., 2018; Genchi et al., 2020), yet still focus on singular series of samples of an isolated area. Missing synoptic analysis bringing together multiple such surveys of varying types and resolutions from

different points in time and space classify most of these data sets as the previously mentioned highly regionalized individually published information. As digital terrain models (DTMs) today already aggregate data and employ spatial interpolation to create high coverage information, the next step is combining spatial and temporal interpolation of multiple data sets to create not only a spatially but also temporally continuous model space for numerous applications.

### 2.1 Introduction to the Functional Seabed Model

The Functional Seabed Model is a holistic data-based hindcast simulation model (Milbradt et al., 2015) and was first utilized in larger scale in joint project AufMod, (Heyer and Schrottke, 2015) to provide consistent DTMs and base data for numerical modelling and morphodynamic analyses from multiple data sets originating from separate surveys for user defined spatial and temporal extent and validity.

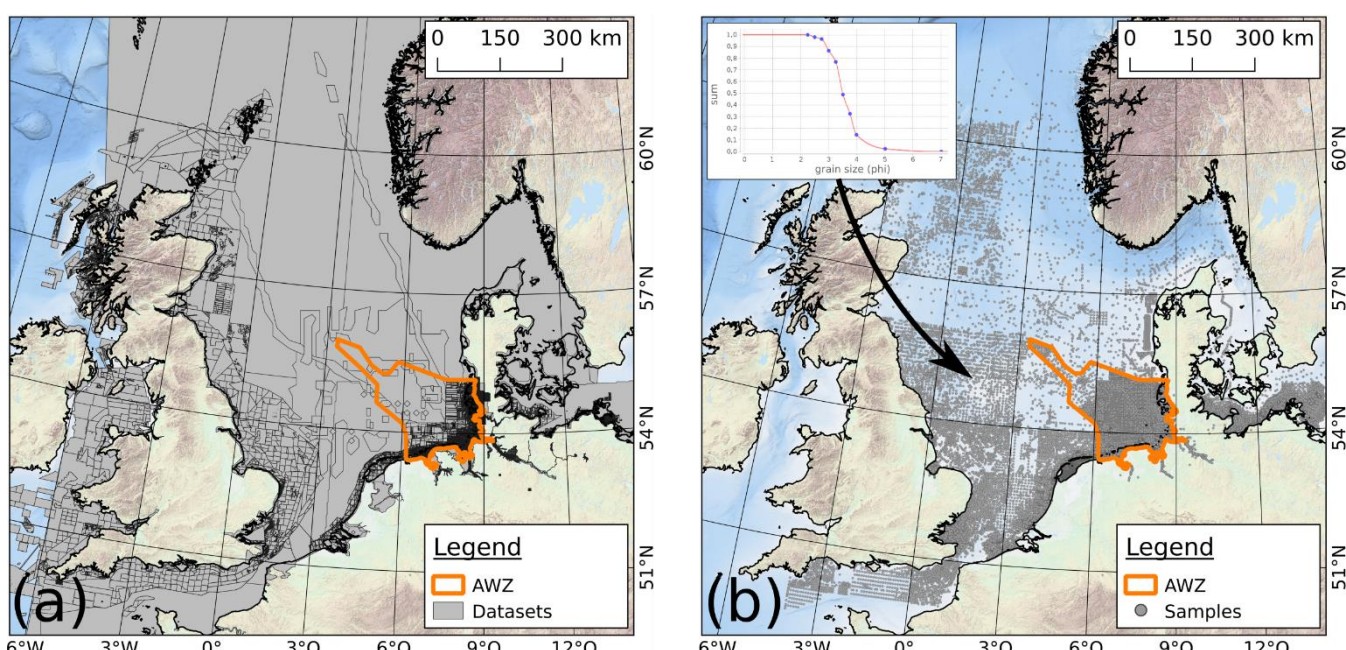

**Figure 2: Spatial extent of FSM database, (a) overlapping bathymetric datasets, (b) surface sedimentological datasets represented by continuous GSD functions.**





In its current form, input data and thus spatial and temporal range cover primarily Germany, the Southern and Central North Sea and the British Isles from the early 1950s until today. Its data base contains approximately 127,000 elevation data sets

with 115 billion single data points (Figure 2a) and approximately 95,000 surface sediment samples represented by continuous grain size distribution (GSD) functions (Figure 2b), to meet requirements of advanced model systems as described in part two of this publication (Hagen et al., 2021, in review). Approximately 45,000 surface sediment samples are located within the AWZ. For the highest attainable transparency of provenance of our products, each individual bathymetric and sedimentological data set hat metadata attached, which include among others sample title, source organization and

survey time as either an isolated instance or a temporal span.

Both bathymetric and surface sedimentological surveys create information at isolated points, thus interpolation and approximation methods are applied to create spatially continuous products for specific points in time. As surveys have varying purposes, they are undertaken as various survey types and a consequence, the structure of the initial processed data set is highly variable as well. Consequently, the anticipated quality of spatial representation is highly dependent on the

applied approximation or interpolation method itself. While linear interpolation may be sufficient in high-density surveys, they might not be adequate in other cases. This is where the FSMs ability to connect geometric data sets to metadata is of utmost importance. The aforementioned metadata information contains not only trivial entries such as the descriptive attributes but also information about the method that is to be utilized for spatial interpolation of the specific data set in question, including parameters such as search radii or required point counts. A small, non-exhaustive example of this survey

type dependent approximation and interpolation definition defined for and utilized in the FSM is displayed in Figure 3.

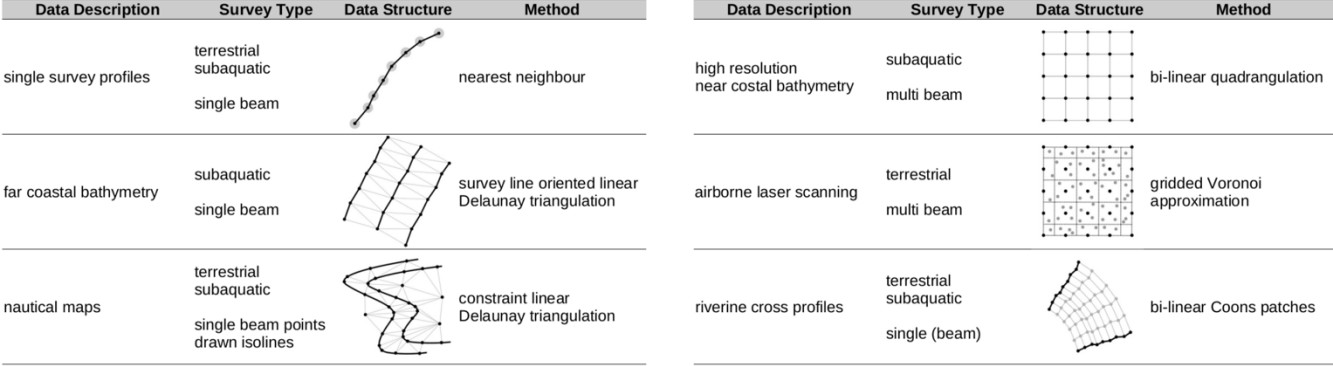

**Figure 3: Overview of the most common approximation and interpolation methods used for datasets of the FSM.**

**2.2 Spatio-Temporal Interpolation and Approximation of Bathymetric Datasets**

Spatial interpolation or approximation of bathymetric data sets in the FSM follows the well-known and fundamental concept of the linear combination of a weighting factor from a so-called base function and the data points elevation information. The



base function is generally dependent on the spatial position of the data point and commonly produces weighting factors based on distances between data points and points to be interpolated or approximated. Both interpolation and approximation

follow therefore Eq. (1), where $\hat{z}(\vec{x})$ is the interpolated or approximated elevation at a position $\vec{x}$, $\varphi_{p^i}(\vec{x})$ is a weighting factor for point $p^i$, which commonly returns 1 at point $p^i$ and 0 for all other points $p^k$ with $i \neq k$, $z_i = z(p^i)$ is the elevation of sample point $p^i$ and $n$ is the number of all data points within the data set.

$$\hat{z}(\vec{x}) = \sum_{i=0}^{n} \varphi_{p^i}(\vec{x}) \cdot z_i \tag{1}$$

While both approximation and interpolation methods rely on the same core concept, only interpolation algorithms, as a special case of approximation, reliably reflect the original data points information at its position. Approximation methods do not need to fulfill this requirement. Regardless of the explicit implementation of the method itself, it is imperative to note that both interpolated and approximated values that do not coincide with the actual data points are estimations. Survey types and consequently interpolation or approximation methods (see Figure 3) define the base functions to be used.

Temporal interpolation between two data points in time at the same spatial position is equivalent to spatial interpolation, but instead utilizes not spatial but temporal distances as parameters for the base functions. It is necessary do employ temporal interpolation as well, as surveys rarely take place at the desired point in time and even if by coincidence they do regionally, on a larger scale it is not reasonably to be expected to have consistent coverage. Due to the generally high density of regional bathymetric surveys spatially and temporally, linear temporal interpolation (compare Eq. (1)) between the closest older and

younger data sets in regard to the point in time and space is determined to be sufficient and subsequently used in the FSM. Combining both spatial and temporal approaches yields the FSMs first and foremost defining quality: The spatio-temporal interpolation. Spatio-temporal interpolation of bathymetric datasets gives the ability to derive an elevation value and consequently DTMs at any given point in space and time within the FSMs model boundaries, see Figure 4.

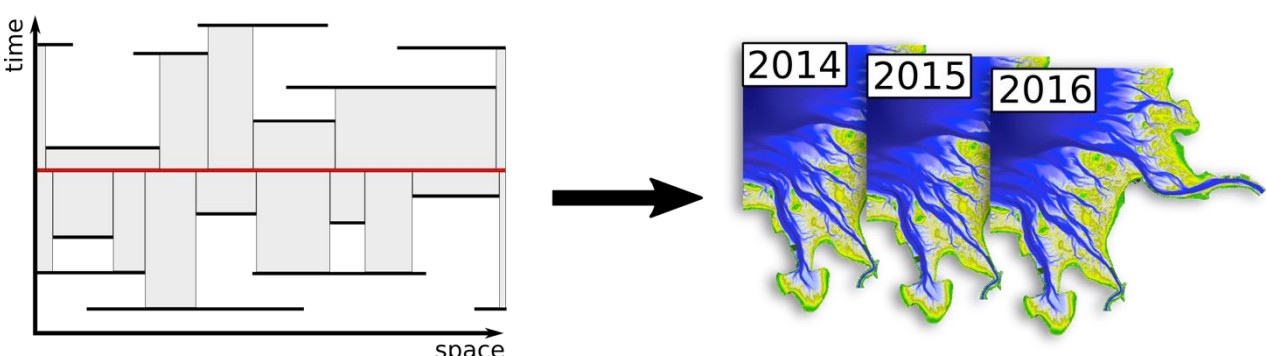


**Figure 4: Concept of spatio-temporal interpolation of bathymetric information.**



This combination is depicted in Eq. (2), where $\hat{z}(\vec{x}, t)$ is the approximated or interpolated value at a position $\vec{x}$ at a time $t$,
$\varphi_i(t)$ is a weighting factor based on the point in time $t$ of data set $i$, which commonly returns 1 at time $t^i$ of data set $i$ and 0

for all $t^k$ with $i \neq k$, and $\hat{z}_i(\vec{x})$ is the interpolated or approximated value at a position $\vec{x}$ within data set $i$, compare Eq. (1),
and $n$ is the number of data sets used.

$$\hat{z}(\vec{x}, t) = \sum_{i=0}^{n} \varphi_i(t) \cdot \hat{z}_i(\vec{x}) \qquad (2)$$

**2.3 Spatio-Temporal Interpolation and Approximation of Surface Sedimentological Datasets**

Spatio-temporal interpolation of surface sedimentological data sets is met with additional obstacles as compared to
bathymetric information. Both the temporal and spatial density of surface sediment data sets, which are comprised of single
sediment samples with single GSD functions, is much lower. Figure 5 displays the temporal distribution of the
approximately 45,000 surface sediment samples within the AWZ (compare Figure 2B).

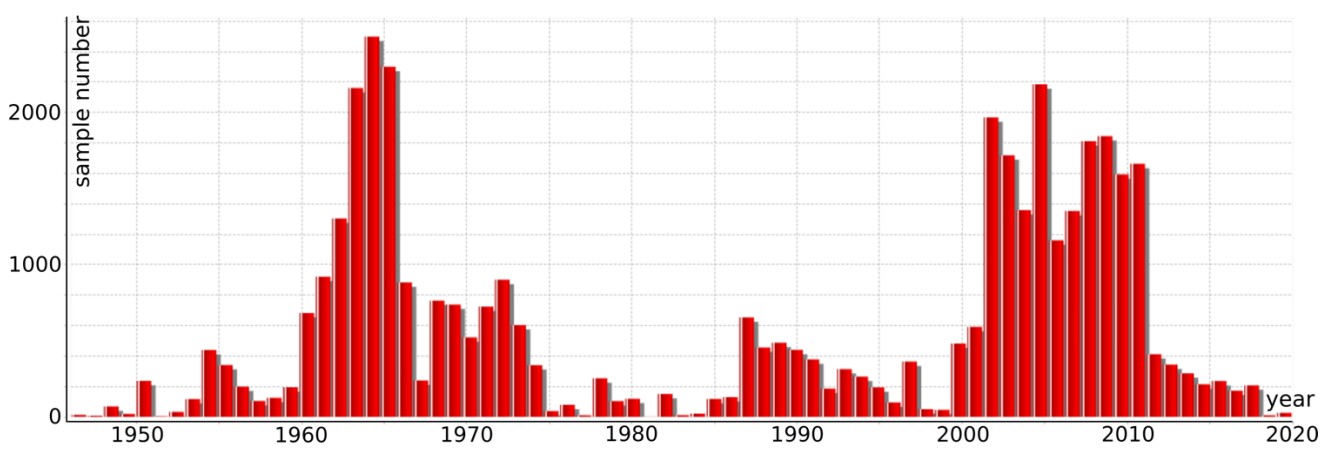


**Figure 5: Temporal distribution of approximately 45,000 surface sediment samples within AWZ.**

Apart from being highly variable, even in one of the years with the most samples taken, 1963, one samples represents on
average approximately 5.7 km$^2$ in the near coastal environment, which is insufficient. Utilizing all samples for the same
point in time, however, would provide approximately one sample in 0.5 km$^2$ for the highly dynamic near coastal

environment, which is in combination with the on average higher spatial density of samples in more active regions,
acceptable. As multiple samples at the same position are virtually non-existent, temporal interpolation as described before
cannot be applied. Thus, an extrapolation approach was developed that relies on the parametrization of the cumulative GSD



function into median grain size, sorting and skewness coefficients after Folk (1980) and an ordinary differential equation as an initial value problem. Equation (3) displays the solution as a time-dependent change $\frac{\partial d_{50}(t)}{\partial t}$ of median grain size $d_{50}$ for a point in time $t$, where $\lambda(z_b(t))$ is a parametrization of depth fuzziness, which reduces influence of elevation changes with greater depths due to measurement uncertainty, $n(t)$ is the time-dependent relative surface porosity (see Sect. 2.3), $\frac{\partial z_b(t)}{\partial t}$ is the step wise sedimentation or erosion rate as derived from all bathymetric data sets at the sample position as a time series, $\sigma_0$ is the initial approximated sorting of the surrounding sediment samples and $\left(1. - \frac{d_{min}}{d_{50}(t)}\right)$ and $\left(1. - \frac{d_{50}(t)}{d_{max}}\right)$ are the logistic boundaries of $d_{50}$. The boundary grain size values $d_{min}$ and $d_{max}$ can be approximated analogous to the initial sorting based on surrounding sediment samples, as $d_{min} = d_5 \cdot 0.5$ and $d_{max} = d_{95} \cdot 2$, respectively.

$$\frac{\partial d_{50}(t)}{\partial t} = \lambda(z_b(t)) \cdot d_{50}(t) \cdot (1 - n(t)) \cdot \frac{\partial z_b(t)}{\partial t} \cdot \sigma_0 \cdot \begin{cases} \left(1. - \dfrac{d_{min}}{d_{50}(t)}\right): & for\ sedimentation\ \dfrac{\partial z_B(t)}{\partial t} > 0 \\ \left(1. - \dfrac{d_{50}(t)}{d_{max}}\right): & for\ erosion\ \dfrac{\partial z_B(t)}{\partial t} \leqslant 0 \end{cases} \tag{3}$$

Based on the change of the median grain size, the change of sorting and skewness coefficients can be approximated as well. With these, the fully continuous cumulative GSD function can be regenerated with Eq. (4), modified after Tauber (1997), as a logistic function, where $\widehat{F(\Phi)}(t)$ is the extrapolated cumulative GSD function depending on grain size in $\Phi$ and extrapolation point in time t, $\Phi_{50}(t)$ is the time-dependent median grain diameter in $\Phi$ resulting from addition of the initial median grain diameter of the original GSD function and the time-dependent change after Eq. (3), $\sigma_I$ is the approximated sorting coefficient and $Sk_I$ is the approximated skewness coefficient (Folk, 1980).

$$\widehat{F(\Phi)}(t) = 1 - \left(1 + e^{-1.7 \cdot (\Phi - \Phi_{50}(t)) \cdot (\sigma_I - Sk_I * \tanh(\Phi - \Phi_{50}(t)))^{-1}}\right)^{-1} \tag{4}$$

The combination of this temporal extrapolation allows for subsequent spatial interpolation or approximation as shown in Eq. (5), where $\widehat{F(\Phi)}(\vec{x}, t)$ is the spatio-temporally interpolated GSD function at position $\vec{x}$ and time $t$, $\widehat{F(\Phi)}_i(t)$ is the temporally extrapolated GSD function at $i$, compare Eq. (4), $\varphi_i(\vec{x})$ is the position dependent weighting factor to $\widehat{F(\Phi)}_i(t)$, which commonly returns 1 at the position of $\widehat{F(\Phi)}_i$ and 0 for all other positions of $\widehat{F(\Phi)}_k$ with $i \neq k$, and $n$ is the number of GSD functions used in the interpolation or approximation method.

$$\widehat{F(\Phi)}(\vec{x}, t) = \sum_{i=0}^{n} \varphi_i(\vec{x}) \cdot \widehat{F(\Phi)}_i(t) \tag{5}$$

Optimized data storage and access solutions were developed, since potentially up to 95,000 references to fully continuous functions had to be handled with acceptable memory usage for each interpolated GSD function in case of global
interpolations without maximum search radii. The weighting factor $\varphi_i(\vec{x})$ is in this application commonly based on Shepard spatial interpolation approaches that is further extended by variable search radii depending on the data density around each specific sample point and hydrodynamic factors such as bed shear stress data, provided by the second part of this publication (Hagen et al., 2021, in review), to introduce anisotropic metrics. Thus, a GSD function can be interpolated at any given point in space and time within the FSMs model boundaries, see Figure 6.


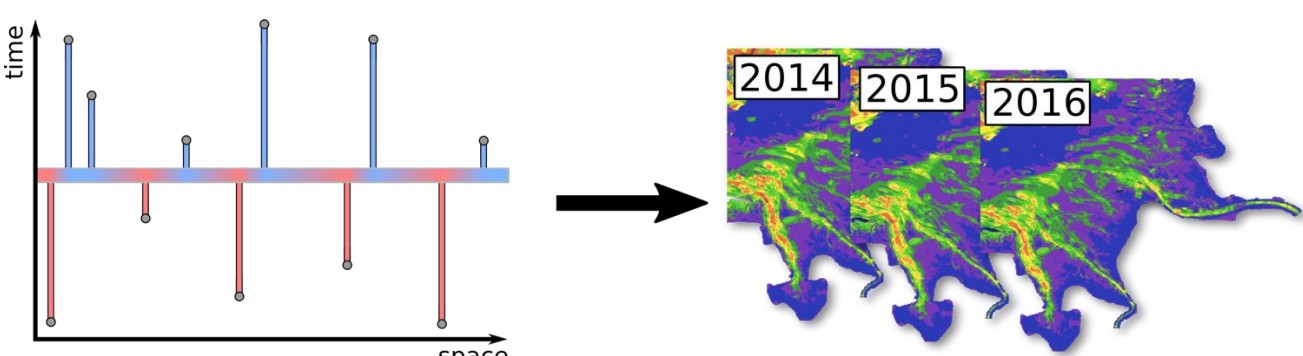

**Figure 6: Concept of spatio-temporal interpolation of surface sedimentological information.**

**2.4 Surface Sediment Porosity**

As utilized in Eq. (3), the surface substrate porosity is an essential factor to consider in development of the sedimentological surface composition of the ocean floor, as generally a lower porosity leads to a denser material that has higher thresholds to erode and consequently leads to less change in elevation and thus after Eq. (3) sedimentological composition. As this is a model approach that is continuously developed and adjusted, we are aware that very well sorted fine materials such as silt and clay actually increase their total porosity, yet as muds in our data base of surface sediment samples tend to be
moderately sorted or lower, we determined this effect to be negligible for the current application. Based on the parameters that can be extracted or extrapolated from the data base, Eq. (6) is modified after Wilson et al. (2018) to adjust porosity $n$ for sorting coefficient $\sigma$ in combination with the median grain size $d_{50}$, where $\mathrm{wc}(d_{50})$ is the settling velocity after Wu and Wang (2006).

$$n = 10^{-0.435+0.366\cdot\left(\frac{1}{1+e^{\left(\frac{-\log_{10}(d_{50})+1.227}{-0.27}\right)}}\right)} \cdot \frac{1}{(1 + \sigma * \mathrm{wc}(d_{50}))} \qquad (6)$$




## 3 Products

### 3.1 Gridded base products

By employing spatial and temporal interpolation as described in Sect. 2, the FSM was used 21 bathymetric digital terrain models (DTM) for the German Bight ranging from 1996 to 2016 with validity dates on 01.07., respectively. Each DTM is

provided as a structured grid with elevation data in a GeoTiff format with a spatial grid resolution of 10 m (EPSG 25832). Furthermore, a 100 m structured grid of the AWZ is provided for 01.07.1996, see Figure 7a.

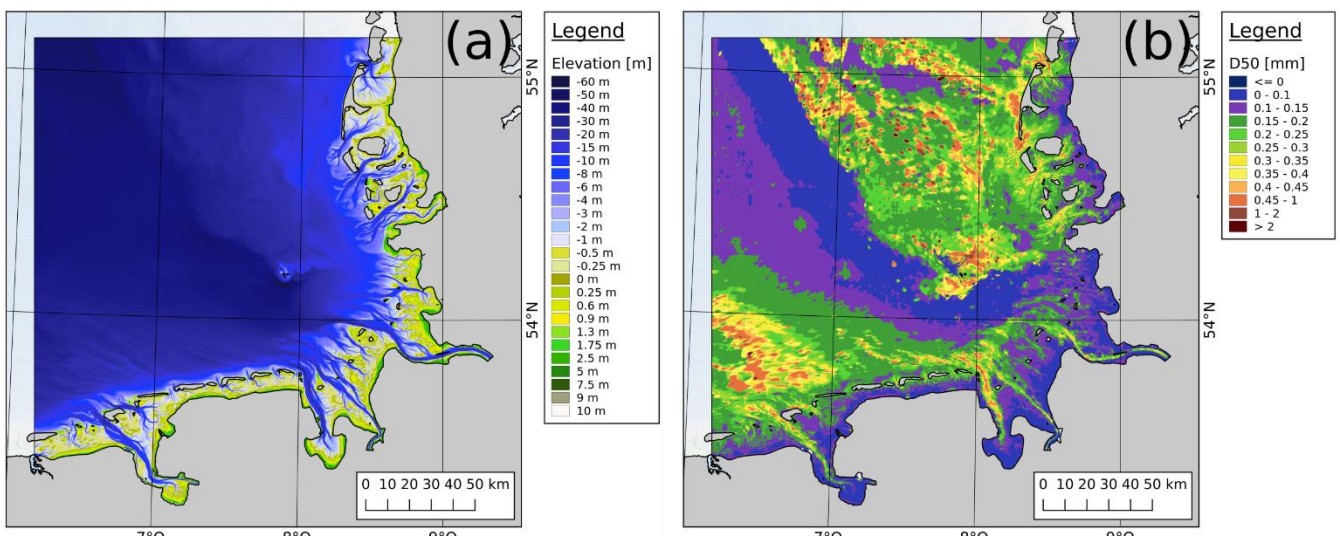

**Figure 7: Exemplary base products for 1996 for (a) bathymetry, (b) surface sedimentology painted as median grain diameter.**


By employing spatial and temporal interpolation and extrapolation as described in Sect. 2, the FSM was used to generate three surface sedimentological models for the German Bight for 1996, 2006 and 2016, with validity dates on 01.07., respectively. Each surface sedimentological model is provided to users serving as a base data set for individual analysis applications as a structured grid as a CSV-file in EPSG 25832 with a spatial grid resolution of 100 m. Each file contains the

model information in a 0.25-phi discretized cumulative GSD function, as well as coordinates and meta data such as date of validity. Furthermore, a 250 m structured grid of the EEZ is provided for 01.07.1996, see Figure 7b represented by a further analysis of the cumulative function (see also later sections): The median grain size $d_{50}$ in mm.

### 3.2 Polygonal base products

Basic bathymetric and surface sedimentological information is in practice often utilized in form of polygonal isoline. We

thus generated isobaths for each bathymetric DTM in full spatial coverage in 0.5 m steps for high resolution analyses and 10 m steps for general display purpose. Additionally, each median grain diameter ($d_{50}$) gridded product is provided with



polygonal representation as well. As the $d_{50}$ is provided in metric scale, a logarithmic discretization for the isolines is required and utilized as defined by the German language version standard DIN EN ISO 14688 (Deutsches Institut für Normung, 2018), which defines grain size fractions on a logarithmic scale.

### 3.3 Gridded analysis products

Based on the DTM products, geomorphological analyses i.e. the development of elevation, minimum and maximum elevation and their range (termed as the bed elevation range (Figure 8a) by Winter, 2011), are performed.

The bed elevation range provides valuable insight into recent morphological activity, as a high value imply strong morphological activity over the analysis time span. The morphological drive (MD, Figure 8b) further expands on the concept of stability by using rates of elevation changes instead of absolute changes, which helps to assess whether an area was affected by gradual (e.g. tidal dynamics) or sudden (e.g. storm surges) change

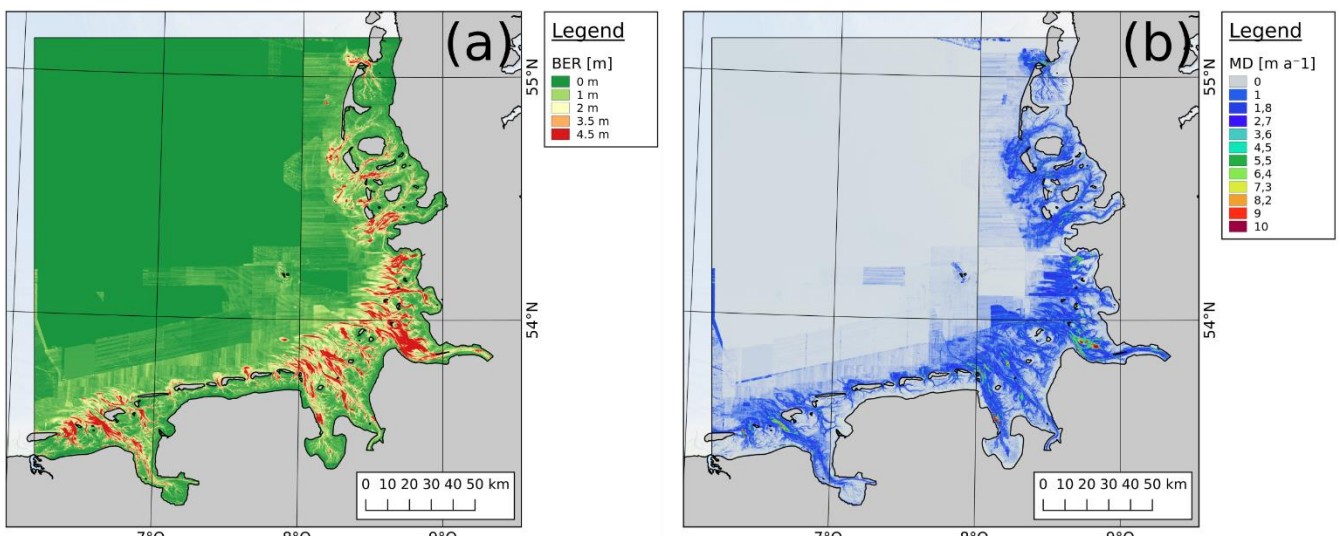

**Figure 8: Bathymetric gridded analysis products, (a) bed elevation range, (b) morphological drive.**

Further sedimentological analyses utilize the full cumulative GSD function from the base products. Based on grids with the same extent and resolution as the base products, calculation of the median grain size $d_{50}$ in mm, the sorting coefficient σ after Folk (1980), the skewness coefficient $Sk$ after Folk (1980) and the porosity $n$, see Eq. (6), were performed (Figure 9).


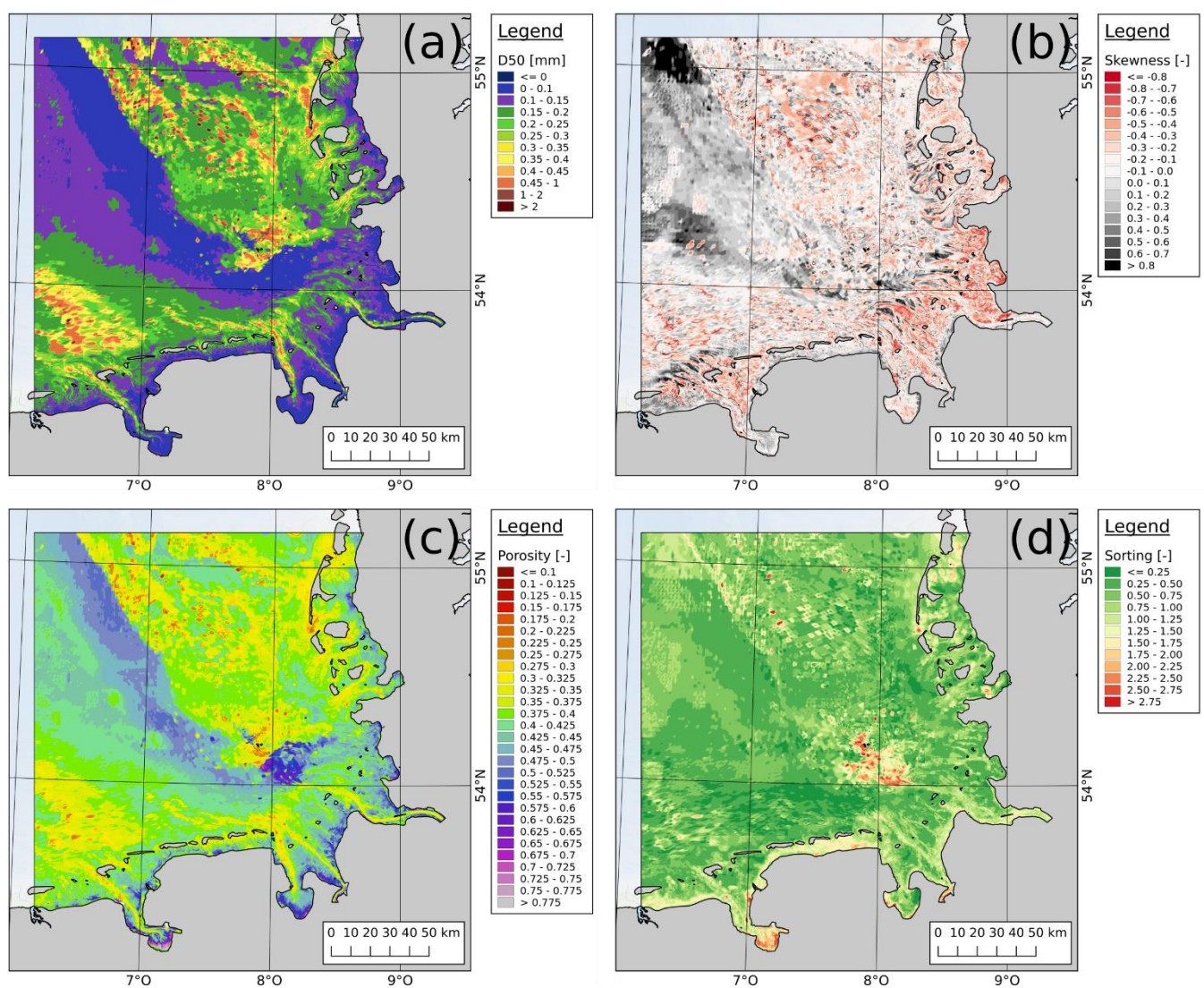

Figure 9: Sedimentological analysis products 1996: (a) median grain size d$_{50}$, (b) skewness, (c) porosity, (d) sorting coefficient.

## 3.4 Polygonal analytical products

For display purposes, sedimentological maps can be produced by a classification of grid points to their linguistic description of their respective cumulative GSD function. The concept of generation of a (quasi-)bijective linguistic description from a function (Sievers et al., in prep.) is based on a reversal of initial concepts from Voss (1982) and Naumann et al. (2014). It relies on the definition of grain size classes and respective percentages within the cumulative function relative to each other to determine a description that would adequately regenerate the original GSD function. With respect to mostly German

stakeholders, the German description format "SEP3" based on grain size classification after standards DIN 4022 and German language version DIN EN ISO 14688 (Deutsches Institut für Normung, 2011, 2018) is used as the target description structure.

While Voss (1982) and Naumann et al. (2014) solely focus on SEP3, the developed reversal of their process could be transferred to multiple other description formats as well. In a similar concept to the Figge grain size classification maps

(Figge, 1981), the description is then split into main and sub components to reduce the number of possible combinations to be displayed. The grid points of these data sets are then summarized into polygonal maps of main and sub component

A major advantage of these so called petrographical maps is that a cumulative GSD function can be reconstructed by recreating a full description from main and sub component descriptions following Naumann et al. (2014) at every single point within the map, without the need to store excessive spatial position data. For each base model, two sets of

petrographical maps are created, one in the long, fully bijective form (i.e. a full description is stored) and a short form (i.e. only the first main and first sub component is stored), which is displayed in Figure 10.

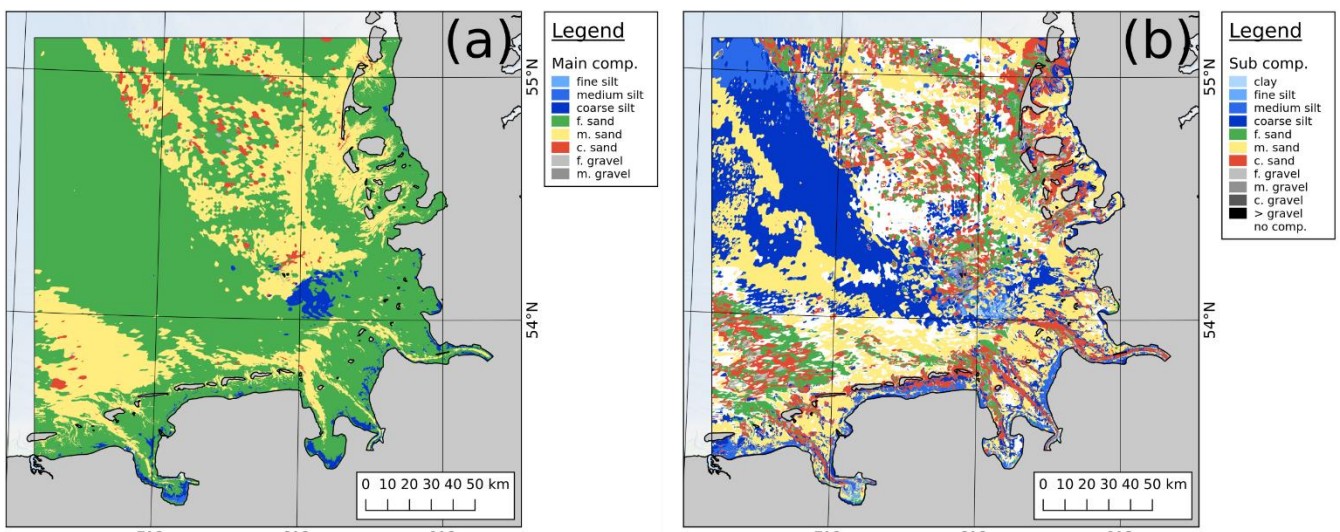

**Figure 10: Petrographical maps after German SEP3 standard, short form 1996: (a) main components, (b) sub component**






## 4 Plausibility Evaluation Products for Gridded Bathymetric Base Products

Commonly, it is not published with a DTM what data sets where used in its creation. A notable exception in this case is EMODnet Bathymetry ([www.emodnet-bathymetry.eu](www.emodnet-bathymetry.eu)), where the possibility to view certain meta data to data sets used for generating the DTM is provided. We aim to make the process of DTM generation fully transparent. As a consequence, each

DTM itself generated by the FSM has specific meta data attached to it, which may give information about possibilities and limitations for certain users and their desired applications. They include common information such as:

1. Dataset title
2. Spatial extent and coordinate reference system, commonly in EPSG notation

3. Elevation range and height reference system, commonly in linguistic notation
4. Short description for potential additional information
5. Interpolation or approximation methods optimal for this data set and possible parameters
6. Data provider and their contact information
7. Date or time span of validity


Additionally, the FSM has the ability to further provide metadata that allow for an evaluation of validity and reliability of the DTM. We supply data density maps and a set of data source maps, which are provided with each single DTM as separate datasets.

### 4.1 Data Density Maps

Data density maps are gridded datasets in identical extent and resolution to its DTM and hold information about the spatial resolution of the underlying spatio-temporally interpolated bathymetric surveys (Figure 11a). The definition of resolution of a field survey is based on their structure, refer to Figure 3. For bi-linear grids it is determined to be the grid cell length, for unstructured datasets with elements it is the mean edge length or elements connected to the point and for unstructured datasets without elements it is the search radius as defined in the interpolation or approximation method (e.g. Shepard-

interpolation). A low value is thus equivalent to a high resolution and is usually correlated with a higher quality, as they tend to be from newer datasets close to or on land and are generally measured with more precise technology with relatively low uncertainty or error, such as airborne laser scanning or multi-beam hydrographic surveys.

### 4.2 Data Source Maps

Data source maps are polygonal datasets which hold the metadata of the survey datasets used for spatio-temporal

interpolation of each individual point inside a DTM. As previously explained, spatio-temporal interpolation in the FSM utilizes two datasets for each point before and after the interpolation date, thus two data source maps for datasets used before





and after the interpolation date, respectively, are provided. To reliably trace provenance of a single elevation value within the DTM, both maps are to be considered. In this context, the provided metadata for each dataset utilized in interpolation are unique identifying IDs within the FSM, the name of the dataset, type and subtype of datasets (implicitly defining interpolation or approximation methods), data source organizations and date or time span of validity, compare Figure 11b.
Especially the date or time span of validity is of high interest, as higher temporal distances to older and younger datasets employed in spatio-temporal interpolation may imply lower reliability for a specific area, compared to other areas in the DTM with lower distances. Data source maps are an especially valuable tool for general quality assurance of a DTM, as potential implausible elevation values in the DTM can be traced back to the original survey datasets, in which a correction of
potential erroneous information may be performed.

The generation of the data source map as such takes place during the spatio-temporal interpolation of the DTM, when each point of the model is individually handled. Attached to the calculation of the elevation itself, the meta data of the older and younger datasets used are stored in a grid of equal size and resolution as the DTM. After the DTM is successfully generated, an algorithm creates boundary polygons for each meta data group, which adhere to common multipolygon logic, in that a
polygon shell can have holes that contain another meta data polygon. The traceability of density, interpolation method and source data set for each individual elevation value of each DTM provides in our belief full transparency regarding potential user applications.

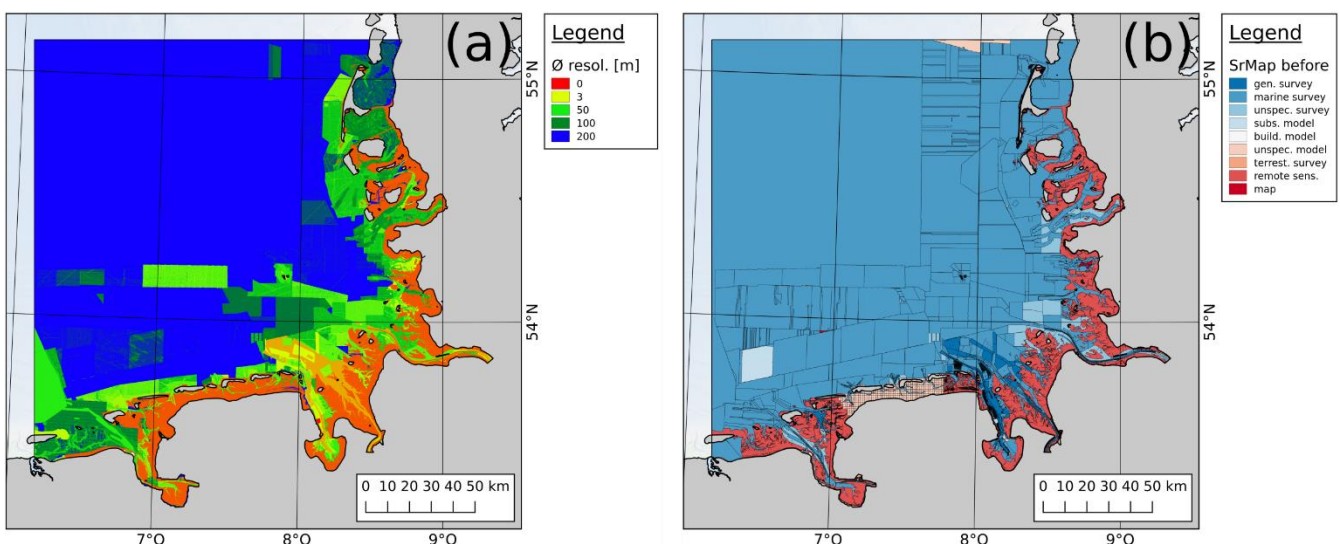

**Figure 11: Plausibility products 2016: (a) data density map, (b) data source map (before) with the survey type as displayed attribute.**



## 5 Data availability

All data sets are open access and available for download as GeoTIFF / ESRI shape files (Geomorphology and bathymetry: [10.48437/02.2020.K2.7000.0001](10.48437/02.2020.K2.7000.0001) Sievers et al., 2020a; Sedimentology: [10.48437/02.2020.K2.7000.0005](10.48437/02.2020.K2.7000.0005), Sievers et al.,
2020b) and additionally as web map (WMS), web feature (WFS) and web coverage services (WCS) for direct integration into a GIS application.

- Geomorphology and bathymetry WMS: [http://mdi-dienste.baw.de/geoserver/EasyGSH_Bathymetrie/wms](http://mdi-dienste.baw.de/geoserver/EasyGSH_Bathymetrie/wms)
- Geomorphology and bathymetry WFS: [http://mdi-dienste.baw.de/geoserver/EasyGSH_Bathymetrie/wfs](http://mdi-dienste.baw.de/geoserver/EasyGSH_Bathymetrie/wfs)
- Geomorphology and bathymetry WCS: [http://mdi-dienste.baw.de/geoserver/EasyGSH_Bathymetrie/wcs](http://mdi-dienste.baw.de/geoserver/EasyGSH_Bathymetrie/wcs)
335       - Surface sedimentology WMS: [http://mdi-dienste.baw.de/geoserver/EasyGSH_Sediment/wms](http://mdi-dienste.baw.de/geoserver/EasyGSH_Sediment/wms)
- Surface sedimentology WFS: [http://mdi-dienste.baw.de/geoserver/EasyGSH_Sediment/wfs](http://mdi-dienste.baw.de/geoserver/EasyGSH_Sediment/wfs)
- Surface sedimentology WCS: [http://mdi-dienste.baw.de/geoserver/EasyGSH_Sediment/wcs](http://mdi-dienste.baw.de/geoserver/EasyGSH_Sediment/wcs)

The project website ([https://mdi-de.baw.de/easygsh/](https://mdi-de.baw.de/easygsh/), last accessed 12th March, 2021) additionally provides a preview web application for the presented data sets.

## 6 Conclusions and Outlook

The produced data are at writing of this publication temporally and spatially the most consistent and continuous available with the highest temporal and spatial resolution and versatility for the German Bight, as no fixed interpretation especially in surface sedimentological data is applied. Numerous new methodologies and validation approaches were developed and implemented, such as the morphological drive and petrographical surface sediment maps. The described Functional Seabed
Model (FSM), as a data-based hindcast simulation model for the bathymetric development of the subaquatic surface and its sedimentological composition, was formed and expanded over a time span of over a decade. As it was designed to be highly modular, it can therefore be expanded with new components very easily. Currently, the integration of the subsurface sediment composition of the morphologically active or activatable space is under development (see methodology excerpt in conference presentation Sievers et al., 2020) in the publicly funded KFKI project "Stratigraphic Model Components for the
Improvement of High-Resolution and Regionalized Morphodynamic Simulation Models" (SMMS) in cooperation with the German Federal Waterways Engineering and Research Institute and the Federal Maritime and Hydrographic Agency of Germany. SMMS aims to provide consistent and continuous stratigraphical data of the sub-seafloor and adjacent estuaries to further improve the ability of hydrodynamic numerical model systems to assess erosional processes. Additionally, ecological model components (see methodology excerpt in conference presentation Rubel et al., 2020) are developed in connection to
the publicly funded KFKI project "Roughness effects of oyster reefs and blue mussel beds in the German Wadden Sea" (BIVA-WATT). With this, we aim to be able to provide the most comprehensive synoptic base and validation data for numerous morphodynamic model systems and other general applications.




# 7 Author Contribution

Julian Sievers – article composition, article figures, article concept, conceptual product design, implementation, product
generation

Peter Milbradt – project initiation, supervising, proof-reading, article concept, conceptual product design, implementation

Romina Ihde – meta data and repository management, digital object identifier registration

Jennifer Valerius – project initiation, proof-reading, base data provision

Robert Hagen – article part I and part II coordination, applicability analyses of products, proof-reading, article concept

Andreas Plüß – project initiation, supervising, applicability analyses of products

# 8 Competing Interests

The authors declare no competing interests.

# 9 Acknowledgements

We thank the German Federal Ministry of Transport (BMVI) for funding the mFUND project EasyGSH-DB (funding no.
19F2004A), which has made this report and every data product presented herein possible. We would like to express their gratitude to all EasyGSH-DB collaborators being Federal Waterways Engineering and Research Institute, Germany (BAW), Küste und Raum and the Federal Maritime and Hydrographic Agency, Germany, (BSH) for their valuable input and enjoyable corroboration. We would also like to acknowledge the many stakeholders, which have provided constant feedback to us and contributed their time to improve our data products presented herein. The bathymetric DTM in form of a Web Map
Service used as background maps in map overview figures has been derived from the EMODnet Bathymetry portal - http://www.emodnet-bathymetry.eu. JS and PM also thank Malte Rubel for discussions regarding sedimentological products and Diego Felipe Pineda Leiva for further proof reading.





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
