# Peer review of "An Integrated Marine Data Collection for the German Bight – Part I: Subaqueous Geomorphology and Surface Sedimentology (1996-2016)"

_Earth System Science Data, 2021_

## Author Response (AR1)

**Referee 1**

**General Comment 1:**

In the introduction, the authors state bathymetric regular 50 m gridded digital terrain models (DTMs) were provided from 1982 to 2012 for the inner German Bight. However, the presented work concerns the bathymetric terrain models spanning 1996 to 2016 as a 10 m regular grid. It should be explained why the authors did not consider the data spanning 1982 to 1996. This should be clarified since this would clearly enrich the numerical modelling results of the second publication, for example.

Response:

The introduction describes the "state of the art" of publicly available data sets including AufMod, which was concluded in 2012, at the beginning of EasyGSH-DB, starting ~2016. This papers granted aim was to generate and provide synoptic bathymetric, morphodynamic and sedimentological data for 20 years (1996 to 2015, compare part II of this publication (Hagen et al, 2021)).
If older data were to be considered, AufMods 50m grid resolution and EasyGSH-DBs 10m grid resolution would produce apparent morphological changes in areas, where small scale structures – like small tidal channels – would appear "closed" due to coarse grid points only on the structures flanks in AufMod but adequately portrayed in EasyGSH-DB. Quick manual-optical analyses regarding this point showed that this could lead to apparent morphological changes of more than 5m, which would automatically invalidate all further analyses and morphodynamical products.
Individual operators of numerical models are, of course, free to utilize both bathymetric and sedimentological data sets from AufMod and EasyGSH-DB, but they have to properly adress the grid-resolution issue on their own accord.

As this paper is, as stated in the abstract (lines 17 and following) and introduction (lines 58 and following), presenting EasyGSH-DB results in its grants constraints and not a "free" morphological analyses study – where further considerations regarding other data sets would be mandatory – we do not see the neccessity for further clarifications in that regard.

Changes:

-

**General Comment 2:**

When considering approximation, interpolation, or extrapolation methods as, for example, for the surface sedimentological data sets, different confidence levels are to be expected. What is the expected accuracy for the given products? This should be clarified by the authors.

Response:

The accuracy of the models, both bathymetric and sedimentological, has two parts.

On the one hand, the accuracy clearly depends on the accuracy of the base data sets. This is usually unknown, as it is not provided in meta data or additional documentation. Typically, the most accurate topographic/bathymetric data sets are generated from airborne laser scanning and have vertical tolerances of +- 15 to 20 cm. Bathymetric data sets like single or multi beam echo sounders are usually estimated to be around 20 cm as well, but this does not consider a) the actual surface that is to be portrayed (top of mud layer or top of solid ground) and b) the fact that uncertainties rise with water depth. As the height of the water column increases, so do the deviations of estimations of travel speeds to real travel speeds of signals. This factor is unknown and/or not provided. Thus, for the sedimentological extrapolation we use an uncertainty estimating factor (lambda). Accuracy of sieving or laser diffraction methods to calculate cumulative functions from sedimentological analyses is unknown and/or not provided.

On the other hand, the accuracy of the model depends as indicated on the methods used. Typically, the bathymetric DTMs use some variant of linear interpolation and thus if a data set is measured at the time stamp of the DTM (01.07.YYYY 00:00:00), the DTM would portray the data set 100% and the accuracy would be identical to the data sets accuracy. As the temporal distance increases, the confidence decreases. The provided data source maps attached to each DTM show for every single grid point which two data sets were used for interpolation and what their time stamps were. Temporal distances can be calculated by the user and used as a measurement of confidence. Sedimentological models were created by anisotropic Shepard interpolation, which guarantees a 100% representation of the sedimentological base data if the point and time of the model coincides with the sample.

In short, generally the accuracy of the models can only be crudely estimated for the bathymetric DTMs, as only bathymetric data sets have in some cases a (at least generalized) scalar accuracy information. Due to the interpolations properties, all data sets will be represented 100% nonetheless provided they are located at a grid point to the models timestamp.

Due to the scarce base information regarding uncertainty of bathymetric measurements and non-existent base information regarding uncertainty of sedimentological analyses, we do not see the possibility to adequately assess model accuracy/uncertainty in the constraints of this paper. Further analyses in these regards would warrant a further paper decoupled from any given grant, where this could be properly adressed.

Changes:

-

Referee 1

**Technical Correction 1:**

Line 99 'data set hat metadata attached'. Please, verify the writing.

Response:

"hat" should be "has".

Changes:

Changed in manuscript

**Technical Correction 2**

Figure 8b: the units given the morphological drive units (MD) appear to be 'm a-1'. Please, check it.

Response:

Morphological drive is calculated as a range between the maximum rate of change and the minimum rate of change and thus has to be m*a^-1.

Changes:

Rephrased explanation of MD in manuscript and further expanded on the concept. Changed unit in legend to m*a^-1.

**Technical Correction 3:**

Line 277: 'were used in its creation' not 'where used in its creation'

Response:

Yes.

Changes:

Changed in revised manuscript.

Referee 1

**Referee 2**

**General Comment 1:**

One minor complaint, at least in the online map viewer, for some variables there were no units visible neither in the legend nor in the metadata (e.g. for the d50.).

Response:

-

Changes:

WMS legends now contain units for sedimentological data (mm for d50, dim.less for porosity, skewness and sorting). ((no changes in manuscript))

**Specific comment 1 (chapter 2.2) – PART 1**

With regards to the temporal interpolation, the authors state that it is sufficient to only use the closest time points in both directions. Have the authors checked if using average changes or trends over a longer time period lead to different results when interpolating the data? If so it might be usefull to go a little bit further into detail here.

Response:

Advanced interpolation methods were evaluated and would have provided a potential benefit in areas where high morphological activity is expected but the data base is scarse. However, highly active areas are also densly sampled in this project area. Regionally, fairways are assessed at least once a month, occasionally even higher frequencies are present. Considering the time frame of EasyGSH-DB, where 20 years are analysed, this would lead to at least ~250 data sets per point in the most frequently sampled regions. To prevent "overshooting" as could be possible in regular polynomial solutions, a bicubic spline interpolation was necessary to investigate further benefitsof advanced interpolation methods. As this spline would need to spatially interpolate elevations on each single data set it utilizes to generate one elevation value at a desired point in time (plus generation of the spline component itself and general database/network traffic), the computation time would increase up to roughly 300 to 400-fold, while there is, due to high temporal sampling density, very little gained. Under the aspect of "cost-benefit"-analysis regarding compational time versus modeled elevation quality, a linear interpolation between the two closest datasets was deemed sufficient. Generally, areas with high morphological activity – where advanced interpolation methods would be useful – are also more frequently sampled, thus decreasing the "cost-benefit"-factor of more complex methods further.

Changes:

Additional explanation concerning "cost-benefit" as reason for not utilizing more advanced interpolations added in chapter 2.2.

Referee 2

**Specific comment 1 (chapter 2.2) – PART 2**

For example in wave dominated areas single extreme storm events can have effects on bathymetric changes that are much larger then during average years. Or the slow movement of large scale bedforms can be observed in bathymetric data while not necessarily indicative of long term erosion/deposition. Allthough these are mostly small scale effects and might not apply to large parts of the data domain, at least mentioning these aspects might help to put the dataset in context.

Response:

EasyGSH-DBs DTMs are only valid for the points in time they are created for, in this case 1st of July of each year between 1996 and 2016. This is a compromise between temporal resolution, data availability, and usability for numerical models. Catastrophic storm events can only be displayed if their influence on bathymetric information used for interpolation is present, e.g. if they happened (shortly) before 1st of July.
If, regionally, there is a suffiecient bathymetric data base, more models, e.g. two per year, four per year or more, could be generated and used in numerical modelling or morphodynamic analyses to accommodate singular extreme events.
To adress long term changes as best as possible, we chose a 20 year period to create the basic models and carry out analyses, as the German Bight and coast line is especially influenced by the 18.6 year cycle of different constellations of Sun, Moon and Earth, which can produce abnormally high or low tides.

Changes:

Additional explanation concerning short term extreme and long term gradual events added to chapter 3.1.

**Specific comment 2 (chapter 2.3)**

The part about the temporal availability of samples (L157-160) is somewhat hard to understand/confusing. For example the authors write "all samples for the same point in time" while to my understanding mean something more like "samples for one point from different times" or even "utilizing all samples regardless of their respective date". This part should be made clearer.

Response:

"utilizing all samples regardless of their respective date" is very concise and makes it much clearer.

Changes:

Rephrasing based on referees suggestion, will be changed in revised manuscript.

Referee 2

**Specific comment 3 (chapter 3.3)**

I assume these products are for the whole period (1996-2016). This should be made clear here again.

Some additional explanation about the calculation of the morphological drive would be helpful. It is not obvious how it helps to differentiate between gradual and sudden changes. Especially since the unit of m per year could also be interpreted as average yearly changes.

Response:

-

Changes:

Specification in the text plus adding a short clarification of what we understand as morphological drive.

**Technical correction 1:**

L 45: that for numerical models --> for use in numerical models

Response:

-

Changes:

Changed in revised manuscript.

**Technical correction 2:**

L 103: and a consequence --> and as a consequence

Response:

-

Changes:

Changed in revised manuscript.

Referee 2

**Technical correction 3:**

L 166: (see Sect. 2.3) --> I think this should be 2.4

Response:
-

Changes:

Changed in revised manuscript.

**Technical correction 4:**

L 213: was used 21 DTMs --> was used for 21 DTMs

Response:
-

Changes:

Changed in revised manuscript.

**Technical correction 5:**

L 229: isoline --> isolines

Response:
-

Changes:

Changed in revised manuscript.

**Technical correction 6:**

L 266: component --> components.

Response:
-

Changes:

Changed in revised manuscript.

Referee 2

**Author's modifications**

-Corrected multiple minor grammatical / syntax errors

-Updated all figures to remove transparencies.

-Added paragraph break in introduction

-Appended temporal range in title to homogenize appearance with Part II and ease user decision whether presented data might be useful without having to read the full abstract